# Comprehensive Online Training and Deployment for Spiking Neural Networks

## Abstract

Spiking Neural Networks (SNNs) are considered to have enormous potential in the future development of Artificial Intelligence (AI) due to their brain-inspired and energy-efficient properties. In the current supervised learning domain of SNNs, compared to vanilla Spatial-Temporal Back-propagation (STBP) training, online training can effectively overcome the risk of GPU memory explosion and has received widespread academic attention. However, the current proposed online training methods cannot tackle the inseparability problem of temporal dependent gradients and merely aim to optimize the training memory, resulting in no performance advantages compared to the STBP training models in the inference phase. To address the aforementioned challenges, we propose Efficient Multi-Precision Firing (EM-PF) model, which is a family of advanced spiking models based on floating-point spikes and binary synaptic weights. We point out that EM-PF model can effectively separate temporal gradients and achieve full-stage optimization towards computation speed and memory footprint. Experimental results have demonstrated that EM-PF model can be flexibly combined with various techniques including random back-propagation, parallel computation and channel attention mechanism, to achieve state-of-the-art performance with extremely low computational overhead in the field of online learning.

## 1 Introduction

Spiking Neural Networks (SNNs), as the third-generation neural network towards brain-inspired intelligence (Maass, 1997), have gained widespread attention from researchers in Artificial Intelligence (AI) community. SNNs utilize spiking neurons as the basic computing unit to transmit discrete spike firing sequences to the postsynaptic layer. Due to the fact that spiking neurons only emit spikes when the membrane potential exceeds the firing threshold, compared to the activation values in traditional Artificial Neural Networks (ANNs), spike sequences have sparse and event-driven properties, which can demonstrate superior computational efficiency and power consumption ratio on neuromorphic hardware (Merolla et al., 2014; Davies et al., 2018; Pei et al., 2019).

Spatial-Temporal Back-propagation (STBP) is the most significant training algorithm in the supervised learning domain of SNNs currently (Wu et al., 2018). By introducing the concepts of temporal dimension and surrogate gradient, STBP simultaneously tackles the Markov property and non-differentiable problem of SNNs existed in the forward propagation and firing process. However, although STBP training has significantly improved the learning performance and universal property of SNNs (Wang et al., 2023; Qiu et al., 2024; Shi et al., 2024), as its back-propagation chains are inseparable due to the temporal dependencies, its GPU memory will inevitably boost linearly with the number of time-steps. This phenomenon greatly increases the training burden and hinders the further application of SNNs to complex scenarios (Kim et al., 2020) and advanced spiking models (Hao et al., 2024).

To address this problem, researchers have transferred the idea of online learning to the STBP training framework (Xiao et al., 2022; Meng et al., 2023), which means that by detaching the temporal dependent gradient terms, SNNs can immediately perform back-propagation at any time-step. This scheme ensures that the corresponding GPU memory is independent of the training time-steps and remains constant, effectively alleviating the problem of computation memory explosion. However,

the current proposed methods based on online learning still have two main defects: (i) the discrepancy between forward and backward propagation, (ii) ineffective online deployment.

The reason for the first defect is that the surrogate function of spiking neurons is generally related to the value of membrane potential and the spike sequence is usually unevenly distributed in the temporal dimension, making the temporal dependent gradients different from each other and unable to merge with the back-propagation chain along the spatial dimension. In this case, when online learning frameworks detach temporal dependent gradients, it will lead to inconsistency between forward and backward propagation, resulting in learning performance degradation. The second defect refers to the fact that current online learning methods mainly focus on optimizing training memory, but cannot bring any optimization regarding computation time or memory during the inference phase. This is because under the framework of firing binary spikes, it is difficult to introduce parallel computation or weight quantization techniques to improve inference speed or optimize memory usage without sacrificing learning precision.

Based on the above discussion, we propose Efficient Multi-Precision Firing (EM-PF) model for online training, it adopts a learning framework with inverted numerical precision, which combines floating-point spikes with binary synaptic weights. On the one hand, EM-PF model solves the non-differentiable problem of the firing process and enhances the uniformity of the spike sequence, significantly improving the separability of the backward gradients compared to vanilla spiking models. On the other hand, the EM-PF model can be flexibly combined with various techniques for optimizing computational costs, achieving full-stage optimization including training and inference phases. Our contributions are summarized as follow:

- Compared to vanilla spiking models, we theoretically point out that EM-PF model has more superior backward gradient separability, which is conducive to achieving high-performance online learning.

- We further propose variant versions based on vanilla EM-PF model. Among them, the EM-PF model based on membrane potential batch-normalization can more effectively regulate the degree of gradient separability and be reparameterized into vanilla EM-PF model in the inference phase. In addition, it can further improve the network performance by combining channel attention mechanism.

- By combining random back-propagation, parallel computation and other techniques, EM-PF model can achieve comprehensive optimization in terms of time and memory overhead during training and inference phases, which goes beyond the optimization scope of vanilla online training.

- We achieve state-of-the-art (SoTA) performance on various datasets with different data-scale and data-type. For example, we reach top-1 accuracy of 79.91% on CIFAR-100 dataset under the condition of saving $15\times$ parameter memory.

## 2 RELATED WORKS

**Recurrent learning algorithms for SNNs.** Considering the similarity in computational mechanisms between SNNs and Recurrent Neural Networks (RNNs), Wu et al. (2018) and Neftci et al. (2019) transferred the Back-propagation Through Time (BPTT) method from RNNs to the supervised learning field of SNNs and utilized surrogate functions to tackle the non-differentiable problem existed in the spike firing process, which is called the STBP training algorithm. On this basis, Li et al. (2021), Guo et al. (2022b) and Wang et al. (2023) respectively attempted to start from the perspective of regulating the distribution about the backward gradient and membrane potential, introducing progressive surrogate functions and penalty terms. Deng et al. (2022) proposed a target learning function which comprehensively considers the SNN output distribution within each time-step, which is particularly suitable for neuromorphic sequential data. To further improve the learning stability and performance of SNNs, various BatchNorm (BN) modules (Zheng et al., 2021; Duan et al., 2022; Guo et al., 2023) and attention mechanisms (Yao et al., 2023; Qiu et al., 2024) have been proposed successively, which capture the representation information contained in spike sequences from multiple dimensions, including spatial-wise, temporal-wise and channel-wise. Recently, advanced spiking models have become a focus of academic attention. Researchers have proposed a variety of neuron models with stronger dynamic properties and memory capabilities

around membrane-related parameters (Fang et al., 2021), firing mechanism (Yao et al., 2022) and dendrite structure (Hao et al., 2024), promoting deeper exploration towards brain-inspired intelligence. In addition, a spatial-temporal back-propagation algorithm based on spike firing time (Bohte et al., 2002; Zhang & Li, 2020; Zhu et al., 2023) has also attracted widespread attention. However, this series of methods are currently limited by high computational complexity and unstable training process, which cannot be effectively applied to complex network backbones and large-scale datasets.

**Online learning algorithm for SNNs.** Although STBP learning algorithm promotes SNNs to join the club of high-performance models, it also brings severe computational burden to SNNs during the training phase, especially the GPU memory that will increase linearly with the number of time-steps. Xiao et al. (2022) transferred the idea of online learning to the domain of SNN direct training, which splits the back-propagation chain by ignoring the backward gradients with temporal dependencies, making the training GPU memory independent of time-steps. On this basis, Meng et al. (2023) proposed a selective back-propagation scheme based on online learning, which significantly improves training efficiency. Yang et al. (2022) combined online learning with ANN-SNN knowledge distillation, further accelerating the training convergence speed of SNNs. Zhu et al. (2024) proposed a brand-new BatchNorm module suitable for online learning, which enhances the stability of gradient calculation by considering the global mean and standard deviation in the temporal dimension. To enrich the neurodynamic property of online learning, Jiang et al. (2024) introduced the difference of membrane potential between adjacent time-steps as a feature term into the backward gradient calculation. Inspired by the architecture of reversed network, Zhang & Zhang (2024) and Hu et al. (2024a) respectively proposed reversible memory-efficient training algorithms from spatial and temporal perspectives. This type of algorithm can ensure computational consistency between online and STBP learning under the condition of occupying constant GPU memory, but it requires bi-directional computation towards all intermediate variables, which inevitably increases computational overhead. In addition, it is worth noting that most of the online learning methods mentioned above neglect the optimization of computation time and memory usage during the inference phase, thus failing to demonstrate their advantages over STBP training when deploying SNN models.

## 3 PRELIMINARIES

**Leaky Integrate and Fire (LIF) model.** The current mainstream spiking model used in SNN community is LIF model, which involves three calculation processes, including charging, firing and resetting. As shown in Eq.(1), at each time-step, LIF model will receive the input current $\boldsymbol{I}_{\mathtt{LIF}}^{l}[t]$ and refer to the previous residual potential $\boldsymbol{v}_{\mathtt{LIF}}^{l}[t-1]$, then accumulate the corresponding membrane potential $\boldsymbol{m}_{\mathtt{LIF}}^{l}[t]$. When $\boldsymbol{m}_{\mathtt{LIF}}^{l}[t]$ has exceeded the firing threshold $\theta^{l}$, a binary spike $\boldsymbol{s}_{\mathtt{LIF}}^{l}[t]$ will be transmitted to the post-synaptic layer and $\boldsymbol{m}_{\mathtt{LIF}}^{l}[t]$ will be reset. Here $\boldsymbol{W}_{\mathtt{float}}^{l}$ denotes the synaptic weight with floating-point precision and $\lambda^{l}$ represents the membrane leakage parameter.

$$\boldsymbol{m}_{\mathtt{LIF}}^{l}[t] = \lambda^{l} \odot \boldsymbol{v}_{\mathtt{LIF}}^{l}[t-1] + \boldsymbol{I}_{\mathtt{LIF}}^{l}[t], \quad \boldsymbol{v}_{\mathtt{LIF}}^{l}[t] = \boldsymbol{m}_{\mathtt{LIF}}^{l}[t] - \boldsymbol{s}_{\mathtt{LIF}}^{l}[t],$$

$$\boldsymbol{I}_{\mathtt{LIF}}^{l}[t] = \boldsymbol{W}_{\mathtt{float}}^{l} \boldsymbol{s}_{\mathtt{LIF}}^{l-1}[t], \quad \boldsymbol{s}_{\mathtt{LIF}}^{l}[t] = \begin{cases} 1, & \boldsymbol{m}_{\mathtt{LIF}}^{l}[t] \geq \theta^{l} \\ 0, & \text{otherwise} \end{cases}. \tag{1}$$

**STBP Training.** To effectively train LIF model, the back-propagation procedure of SNNs usually chooses to expand along both spatial and temporal dimensions, as shown in Fig.1(a). We use $\mathcal{L}$ to denote the target loss function. As shown in Eq.(2), $\frac{\partial \mathcal{L}}{\partial \boldsymbol{m}_{\mathtt{LIF}}^{l}[t]}$ depends on both $\frac{\partial \mathcal{L}}{\partial \boldsymbol{s}_{\mathtt{LIF}}^{l}[t]}$ and $\frac{\partial \mathcal{L}}{\partial \boldsymbol{m}_{\mathtt{LIF}}^{l}[t+1]}$ simultaneously, while the non-differentiable problem of $\frac{\partial \boldsymbol{s}_{\mathtt{LIF}}^{l}[t]}{\partial \boldsymbol{m}_{\mathtt{LIF}}^{l}[t]}$ will be tackled through calculating approximate surrogate functions. Although STBP training enables SNN to achieve relatively superior performance, it inevitably causes severe memory overhead during the training process, which will increase linearly with the number of time-steps.

$$\frac{\partial \mathcal{L}}{\partial \boldsymbol{m}_{\mathtt{LIF}}^{l}[t]} = \underbrace{\frac{\partial \mathcal{L}}{\partial \boldsymbol{s}_{\mathtt{LIF}}^{l}[t]} \frac{\partial \boldsymbol{s}_{\mathtt{LIF}}^{l}[t]}{\partial \boldsymbol{m}_{\mathtt{LIF}}^{l}[t]}}_{\text{spatial dimension}} + \underbrace{\frac{\partial \mathcal{L}}{\partial \boldsymbol{m}_{\mathtt{LIF}}^{l}[t+1]} \frac{\partial \boldsymbol{m}_{\mathtt{LIF}}^{l}[t+1]}{\partial \boldsymbol{m}_{\mathtt{LIF}}^{l}[t]}}_{\text{temporal dimension}}.$$

$$\nabla_{\boldsymbol{W}_{\mathtt{float}}^{l}} \mathcal{L} = \sum_{t=1}^{T} \frac{\partial \mathcal{L}}{\partial \boldsymbol{m}_{\mathtt{LIF}}^{l}[t]} \frac{\partial \boldsymbol{m}_{\mathtt{LIF}}^{l}[t]}{\partial \boldsymbol{W}_{\mathtt{float}}^{l}}, \quad \frac{\partial \boldsymbol{m}_{\mathtt{LIF}}^{l}[t+1]}{\partial \boldsymbol{m}_{\mathtt{LIF}}^{l}[t]} = \lambda^{l} + \frac{\partial \boldsymbol{m}_{\mathtt{LIF}}^{l}[t+1]}{\partial \boldsymbol{s}_{\mathtt{LIF}}^{l}[t]} \frac{\partial \boldsymbol{s}_{\mathtt{LIF}}^{l}[t]}{\partial \boldsymbol{m}_{\mathtt{LIF}}^{l}[t]}. \tag{2}$$

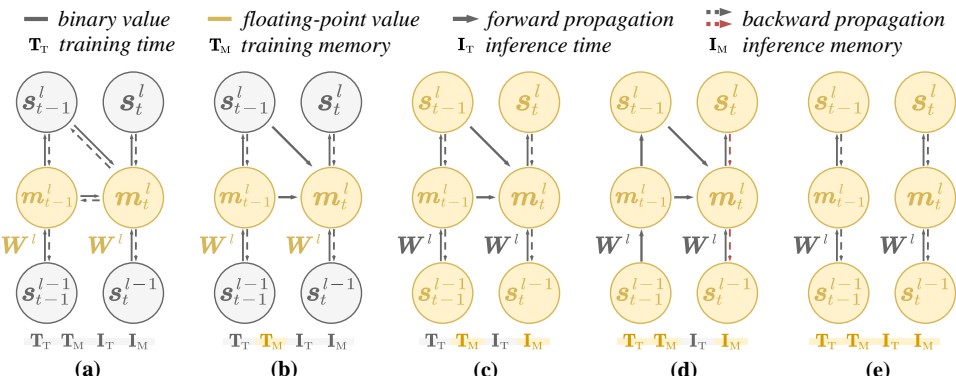

Figure 1: Various training frameworks for SNNs in synaptic and neuron layers. (a): STBP training, (b): vanilla online training based on LIF model, (c)-(e): online training, random back-propagation and parallel computation based on EM-PF model.

**Online Training.** To avoid the issue of training memory overhead, a feasible solution is to ignore $\frac{\partial \mathcal{L}}{\partial \boldsymbol{m}^l_{\text{LIF}}[t+1]} \frac{\partial \boldsymbol{m}^l_{\text{LIF}}[t+1]}{\partial \boldsymbol{m}^l_{\text{LIF}}[t]}$ during the gradient calculation process, thereby making the back-propagation chain independent in the temporal dimension, as illustrated in Fig.1(b). Online training enables SNNs to update gradients at any time-step, keeping the GPU memory at a constant level.

## 4 METHODS

### 4.1 OVERCOMING THE BACK-PROPAGATION DISCREPANCY OF ONLINE TRAINING

From Eq.(2), one can find that the backward gradient of STBP training can also be rewritten as $\frac{\partial \mathcal{L}}{\partial \boldsymbol{m}^l_{\text{LIF}}[t]} = \sum_{i=t}^{T} \frac{\partial \mathcal{L}}{\partial \boldsymbol{s}^l_{\text{LIF}}[i]} \frac{\partial \boldsymbol{s}^l_{\text{LIF}}[i]}{\partial \boldsymbol{m}^l_{\text{LIF}}[i]} \prod_{j=t+1}^{i} \frac{\partial \boldsymbol{m}^l_{\text{LIF}}[j]}{\partial \boldsymbol{m}^l_{\text{LIF}}[j-1]}$. On this basis, we point out the concept of Separable Backward Gradient:

**Definition 4.1.** When $\frac{\partial \mathcal{L}}{\partial \boldsymbol{s}^l_{LIF}[1]} = ... = \frac{\partial \mathcal{L}}{\partial \boldsymbol{s}^l_{LIF}[T]}$, if the surrogate function of $\boldsymbol{s}^l_{LIF}[t]$ w.r.t. $\boldsymbol{m}^l_{LIF}[t]$ is constant, we will have $\frac{\partial \mathcal{L}}{\partial \boldsymbol{m}^l_{LIF}[t]} = \frac{\partial \mathcal{L}}{\partial \boldsymbol{s}^l_{LIF}[t]} \sum_{i=t}^{T} \boldsymbol{\epsilon}^l[i,t]$, here $\boldsymbol{\epsilon}^l[i,t] = \frac{\partial \boldsymbol{s}^l_{LIF}[i]}{\partial \boldsymbol{m}^l_{LIF}[i]} \prod_{j=t+1}^{i} \frac{\partial \boldsymbol{m}^l_{LIF}[j]}{\partial \boldsymbol{m}^l_{LIF}[j-1]}$ denotes the temporal gradient contribution weight of the i-th step w.r.t. the t-th step, which is a constant value. Therefore, we can further have $\left( \frac{\partial \mathcal{L}}{\partial \boldsymbol{m}^l_{LIF}[t]} \right)_{Online} \Leftrightarrow \left( \frac{\partial \mathcal{L}}{\partial \boldsymbol{m}^l_{LIF}[t]} \right)_{STBP}$, the gradient at this point is called **Separable Backward Gradient**.

When the precondition of Definition 4.1 holds true, the back-propagation chain can be considered separable in the temporal dimension and the backward gradient of online training can be seamlessly transformed from that of STBP training. Unfortunately, vanilla STBP training generally requires surrogate gradient functions which are related to the membrane potential value (*e.g.* $\frac{\partial \boldsymbol{s}^l_{\text{LIF}}[t]}{\partial \boldsymbol{m}^l_{\text{LIF}}[t]} = \frac{1}{\theta^l} \max\left( \theta^l - |\boldsymbol{m}^l_{\text{LIF}}[t] - \theta^l|, 0 \right)$), to provide richer information for binary spikes with limited representation capabilities. Therefore, current online training cannot fully overcome the discrepancy between forward and backward propagation, which also limits its learning precision.

To tackle this problem, we propose the EM-PF model, which is an advanced spiking model suitable for online training. As shown in Eq.(3) and Fig.1(c), compared to vanilla LIF model, EM-PF model emits spikes $\boldsymbol{s}^l[t]$ with floating-point value to the binary synaptic layers $\boldsymbol{W}^l$ through various activation functions **ActFunc**($\cdot$). For neurons that emit spikes, it is worth noting that the overall firing process is completely differentiable and corresponding membrane potential will be reset to the position of firing threshold $\theta^l[t]$.

$$\boldsymbol{m}^l[t] = \lambda^l[t] \odot \boldsymbol{v}^l[t-1] + \boldsymbol{I}^l[t], \quad \boldsymbol{v}^l[t] = \boldsymbol{m}^l[t] - \boldsymbol{s}^l[t], \quad \boldsymbol{I}^l[t] = \boldsymbol{W}^l \boldsymbol{s}^{l-1}[t],$$

$$\boldsymbol{W}^l = \texttt{Sign}(\boldsymbol{W}^l_{\texttt{float}}), \; \boldsymbol{s}^l[t] = \texttt{ActFunc}(\boldsymbol{m}^l[t] - \theta^l[t]) = \begin{cases} \boldsymbol{m}^l[t] - \theta^l[t], & \boldsymbol{m}^l[t] \geq \theta^l[t] \\ 0, & \text{otherwise} \end{cases} . \quad (3)$$

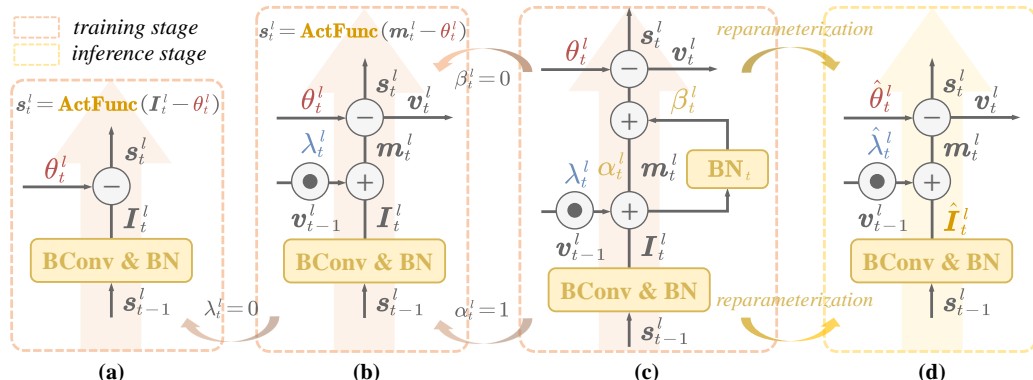

Figure 2: Various versions of EM-PF model. (a): parallel computation, (b): learnable membrane-parameters, (c): membrane potential batch-normalization, (d) the model after reparameterization in the inference stage.

According to Eq.(3), we can derive the back-propagation chain of EM-PF model during the online training process as follow:

$$\nabla_{\boldsymbol{W}^l_{\text{float}}} \mathcal{L} = \sum_{t=1}^{T} \frac{\partial \mathcal{L}}{\partial \boldsymbol{m}^l[t]} \frac{\partial \boldsymbol{m}^l[t]}{\partial \boldsymbol{W}^l} \frac{\partial \boldsymbol{W}^l}{\partial \boldsymbol{W}^l_{\text{float}}}, \quad \frac{\partial \mathcal{L}}{\partial \boldsymbol{m}^l[t]} = \frac{\partial \mathcal{L}}{\partial \boldsymbol{s}^l[t]} \frac{\partial \boldsymbol{s}^l[t]}{\partial \boldsymbol{m}^l[t]}.$$

$$\frac{\partial \boldsymbol{W}^l}{\partial \boldsymbol{W}^l_{\text{float}}} = \begin{cases} 1, & -1 \leq \boldsymbol{W}^l_{\text{float}} \leq 1 \\ 0, & \text{otherwise} \end{cases}, \quad \frac{\partial \boldsymbol{s}^l[t]}{\partial \boldsymbol{m}^l[t]} = \begin{cases} 1, & \boldsymbol{m}^l[t] \geq \theta^l[t] \\ 0, & \text{otherwise} \end{cases}. \tag{4}$$

Here $\boldsymbol{W}^l$ adopts a learning mode of Binary Convolutional (BConv) layer (Liu et al., 2021). We choose ReLU as **ActFunc**$(\cdot)$ to make the surrogate gradient of EM-PF model independent of the corresponding membrane potential value. Considering the unique properties of EM-PF model in terms of spike firing mechanism and surrogate gradient values, we can further propose the following theorem:

**Theorem 4.2.** *In the following two cases, the back-propagation of EM-PF model satisfies the condition of **Separable Backward Gradient** and $\forall i > t, \boldsymbol{\epsilon}^l[i,t] = \boldsymbol{0}$:*
*(i) $\boldsymbol{I}^l[1] \geq \theta^l[1] - \lambda^l[1]\boldsymbol{v}^l[0]$; $\forall t \geq 2, \boldsymbol{I}^l[t] \geq \theta^l[t] - \lambda^l[t]\theta^l[t-1]$.*
*(ii) $\forall t \in [1, T], \boldsymbol{I}^l[t] < \theta^l[t] - \lambda^l[t]\boldsymbol{v}^l[t-1]$.*

Detailed proof is provided in the Appendix. From Theorem 4.2, one can find that EM-PF model can reduce the discrepancy between forward and backward propagation more effectively. On this basis, we set learnable membrane leakage parameters $\lambda^l[t]$ and thresholds $\theta^l[t]$ for EM-PF model at each time-step, enabling EM-PF model to more adaptively regulate the separability of its learning gradient during online training, as shown in Eq.(5) and Fig.2(b).

$$\frac{\partial \mathcal{L}}{\partial \lambda^l[t]} = \frac{\partial \mathcal{L}}{\partial \boldsymbol{m}^l[t]} \frac{\partial \boldsymbol{m}^l[t]}{\partial \lambda^l[t]}, \quad \frac{\partial \mathcal{L}}{\partial \theta^l[t]} = \frac{\partial \mathcal{L}}{\partial \boldsymbol{s}^l[t]} \frac{\partial \boldsymbol{s}^l[t]}{\partial \theta^l[t]}, \quad \frac{\partial \boldsymbol{s}^l[t]}{\partial \theta^l[t]} = \begin{cases} -1, & \boldsymbol{m}^l[t] \geq \theta^l[t] \\ 0, & \text{otherwise} \end{cases}. \tag{5}$$

### 4.2 EM-PF MODEL WITH MEMBRANE POTENTIAL BATCH-NORMALIZATION

In EM-PF model, the distribution of $\boldsymbol{m}^l[t]$ plays a crucial role: on the one hand, it affects the distribution of input current in the post-synaptic layer at the current and subsequent time-steps; on the other hand, it regulates the distribution of surrogate gradients. From Definition 4.1 and Theorem 4.2, we can note that the above two aspects will jointly determine the separability degree of the backward gradient during the online training process, thereby indirectly affecting the learning performance of SNNs. Therefore, based on the vanilla EM-PF model, we propose a novel version that enables to

Table 1: Comparison of computational overhead among various training frameworks during the training and inference phases.

| Method | Random BP | Parallel Comput. | Train. Time | Train. Mem. | Inf. Time | Inf. Mem. |
|---|---|---|---|---|---|---|
| STBP Training | N/A | N/A | ✗ | ✗ | ✗ | ✗ |
| Online Training | ✗ | N/A | ✗ | ✓ | ✗ | ✗ |
| | ✓ | N/A | ✓ | ✓ | ✗ | ✗ |
| **Ours** | ✗ | ✗ | ✗ | ✓ | ✗ | ✓ |
| | ✓ | ✗ | ✓ | ✓ | ✗ | ✓ |
| | ✓ | ✓ | ✓ | ✓ | ✓ | ✓ |

regulate $\boldsymbol{m}^l[t]$ through membrane potential batch-normalization:

$$\boldsymbol{m}^l[t] = \lambda^l[t] \odot \boldsymbol{v}^l[t-1] + \boldsymbol{I}^l[t], \quad \hat{\boldsymbol{m}}^l[t] = \alpha^l[t] \odot \boldsymbol{m}^l[t] + \beta^l[t] \odot \mathtt{BN}_t(\boldsymbol{m}^l[t]),$$

$$\mathtt{BN}_t(\boldsymbol{m}^l[t]) = \gamma \cdot \frac{\boldsymbol{m}^l[t] - \mu_t}{\sqrt{\sigma_t^2 + \epsilon}} + b, \ \boldsymbol{v}^l[t] = \hat{\boldsymbol{m}}^l[t] - \boldsymbol{s}^l[t], \ \boldsymbol{s}^l[t] = \mathtt{ActFunc}(\hat{\boldsymbol{m}}^l[t] - \theta^l[t]). \quad (6)$$

Here $\mu_t$ and $\sigma_t$ are the mean and standard deviation of $\boldsymbol{m}^l[t]$, while $\gamma$ and $\epsilon, b$ are the scaling and shifting factors of the BatchNorm layer. $\alpha^l[t]$ and $\beta^l[t]$ are learnable parameters that regulate the normalization degree of $\boldsymbol{m}^l[t]$. When $\alpha^l[t] = 1, \beta^l[t] = 0$, our model will degrade to the vanilla learnable EM-PF model mentioned in Section 4.1, which ensures the performance lower-bound. As illustrated in Fig.2(c), we assign corresponding $\mathtt{BN}_t(\cdot)$ for each time-step, achieving more precise control for membrane potential distribution and gradient separability. At this point, we can rewrite Theorem 4.2 as follow:

**Corollary 4.3.** *In the following two cases, the back-propagation of EM-PF model (membrane potential batch-normalization version) satisfies the condition of **Separable Backward Gradient**:*
*(i)* $\boldsymbol{I}^l[1] \geq \frac{\sqrt{\sigma_t^2 + \epsilon}\left(\theta^l[1] - \lambda^l[1]\boldsymbol{v}^l[0] + \beta^l[t]b\right) + \gamma\beta^l[t]\mu_t}{\alpha^l[t]\sqrt{\sigma_t^2 + \epsilon} + \gamma\beta^l[t]}; \ \forall t \geq 2, \boldsymbol{I}^l[t] \geq \frac{\sqrt{\sigma_t^2 + \epsilon}\left(\theta^l[t] - \lambda^l[t]\theta^l[t-1] + \beta^l[t]b\right) + \gamma\beta^l[t]\mu_t}{\alpha^l[t]\sqrt{\sigma_t^2 + \epsilon} + \gamma\beta^l[t]}.$
*(ii)* $\forall t \in [1, T], \boldsymbol{I}^l[t] < \frac{\sqrt{\sigma_t^2 + \epsilon}\left(\theta^l[t] - \lambda^l[t]\boldsymbol{v}^l[t-1] + \beta^l[t]b\right) + \gamma\beta^l[t]\mu_t}{\alpha^l[t]\sqrt{\sigma_t^2 + \epsilon} + \gamma\beta^l[t]}.$

In addition, it is worth noting that the EM-PF model based on membrane potential batch-normalization can also be converted into vanilla learnable EM-PF model through reparameterization during the inference phase, thereby avoiding the introduction of additional computational overhead, as shown in Fig.2(c)-(d). Firstly, we can integrate Eq.(6) into the following equation:

$$\hat{\boldsymbol{m}}^l[t] = (\alpha^l[t] + \frac{\gamma \cdot \beta^l[t]}{\sqrt{\sigma_t^2 + \epsilon}})\boldsymbol{m}^l[t] - \beta^l[t](\frac{\gamma \cdot \mu_t}{\sqrt{\sigma_t^2 + \epsilon}} + b). \quad (7)$$

Subsequently, we can merge the scaling and shifting terms *w.r.t.* $\boldsymbol{m}^l[t]$ in Eq.(7) into membrane-related parameters at different positions, including membrane leakage parameters, threshold, and input current:

$$\hat{\lambda}^l[t] = (\alpha^l[t] + \frac{\gamma \cdot \beta^l[t]}{\sqrt{\sigma_t^2 + \epsilon}})\lambda^l[t], \quad \hat{\theta}^l[t] = \theta^l[t] + \beta^l[t](\frac{\gamma \cdot \mu_t}{\sqrt{\sigma_t^2 + \epsilon}} + b),$$

$$\hat{\boldsymbol{I}}^l[t] = (\alpha^l[t] + \frac{\gamma \cdot \beta^l[t]}{\sqrt{\sigma_t^2 + \epsilon}})\boldsymbol{I}^l[t], \quad \boldsymbol{v}^l[t] = \boldsymbol{m}^l[t] - \boldsymbol{s}^l[t] - \beta^l[t](\frac{\gamma \cdot \mu_t}{\sqrt{\sigma_t^2 + \epsilon}} + b). \quad (8)$$

### 4.3 STRENGTHENING THE PERFORMANCE OF ONLINE LEARNING AND DEPLOYMENT

As illustrated in Fig.1(a)-(b) and Tab.1, compared to STBP training, vanilla online training only saves GPU memory during the training phase, without providing any advantages in terms of computation time or memory overhead during the inference phase, which impedes effective online deployment for trained models. In comparison, we point out that online training based on EM-PF model can achieve full-stage computational optimization:

- **Training time**: As shown in Fig.1(d), we transfer the idea of random back-propagation (Meng et al., 2023) to the online training of EM-PF model, which means that we randomly

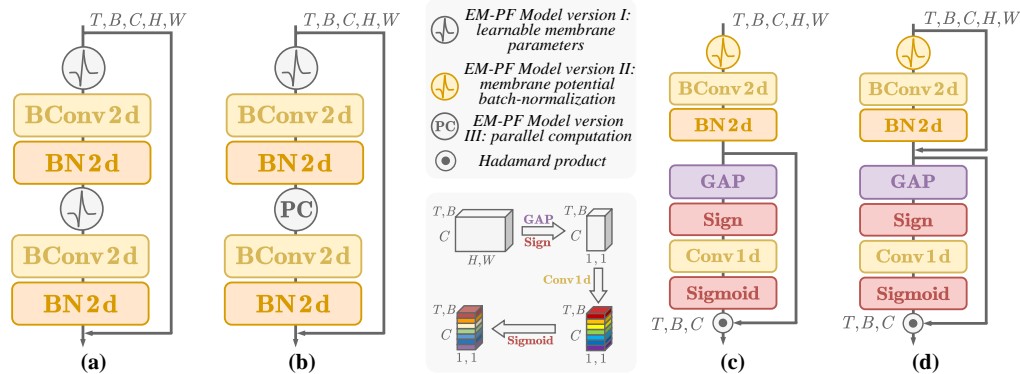

Figure 3: ResNet blocks based on different versions of EM-PF model and SECA modules. (a): vanilla block, (b): parallel acceleration block, (c)-(d): blocks based on channel attention mechanism.

select only 1 time-step within $T$ time-steps for each mini-batch data to propagate backward gradient. This technique significantly improves the back-propagation efficiency and reduces the gradient computation load from $O(T)$ to $O(1)$.

- **Training memory**: Consistent with previous online training methods, EM-PF model does not consider the computation of $\frac{\partial \boldsymbol{m}^l[t+1]}{\partial \boldsymbol{m}^l[t]}$, ensuring that the GPU memory is independent of the number of training time-steps and keeps constant.

- **Inference time**: We introduce a parallel computation version for EM-PF model to accelerate inference time, as shown in Fig.1(e) and Fig.2(a). This simplified EM-PF model does not take into account residual membrane potential information from previous time-steps, allowing for simultaneous forward calculation for all time-steps. In addition, it does not require $\boldsymbol{v}^l[t-1]$ to be kept as an intermediate variable during the training process (to update learnable $\lambda^l[t]$), which further saves training GPU memory.

- **Inference memory**: As EM-PF model emits spikes with floating-point type, to maintain the computational principles of SNNs, we conduct binary training and storage for synaptic layers, which makes our model merely occupy minimal memory and become more suitable for online deployment.

### 4.4 ENHANCING ONLINE TRAINING THROUGH CHANNEL ATTENTION MECHANISM

Channel Attention mechanism (Hu et al., 2018; Wang et al., 2020; Guo et al., 2022a) is usually inserted after the convolutional layers to optimize the network performance. We transfer the idea of ECA (Wang et al., 2020) to the online training based on EM-PF model, then propose Spiking Efficient Channel Attention (SECA) mechanism, as shown in Eq.(9).

$$\mathtt{SECA}(\boldsymbol{I}^l[t]) = \mathtt{Sigmoid}(\underbrace{\mathtt{Conv1d}}_{\in \mathbb{R}^{1 \times 1 \times K}}(\underbrace{\mathtt{Sign}(\mathtt{GAP}(\boldsymbol{I}^l[t]))}_{\in \mathbb{R}^{B \times 1 \times C}})) \odot \boldsymbol{I}^l[t], \ \boldsymbol{I}^l[t] \in \mathbb{R}^{B \times C \times H \times W}. \quad (9)$$

Here the input current $\boldsymbol{I}^l[t] \in \mathbb{R}^{B \times C \times H \times W}$ will be compressed to $\mathbb{R}^{B \times C \times 1 \times 1}$ through the Global Average Pooling (GAP) layer, then $\mathtt{Conv1d}(\cdot)$ and $\mathtt{Sigmoid}(\cdot)$ will be used to capture and activate the attention scores among different channels, ultimately merging with the shortcut path. Considering that the EM-PF model can convey enough information representation at each time-step, we enable the spike sequence to share the weight of SECA in the temporal dimension. Due to its extremely low parameter quantity (only 1 Conv1d layer with $1 \times 1 \times K$ parameters), SECA can further enhance the learning ability of SNNs under the condition of hardly affecting its online deployment.

In addition, as shown in Eq.(10) and Fig.3(c)-(d), we further propose two variants for SECA:

$$\mathtt{SECA\text{-}I}(\boldsymbol{I}^l[t]) : \mathtt{SECA}(\boldsymbol{I}^l[t]), \ \ \mathtt{SECA\text{-}II}(\boldsymbol{I}^l[t]) : \mathtt{SECA}(\boldsymbol{I}^l[t] + \mathtt{BN2d}(\mathtt{BConv2d}(\boldsymbol{I}^l[t]))). \quad (10)$$

Here, $\mathtt{SECA\text{-}I}(\cdot)$ is the conventional channel attention mechanism, while considering the shortcomings of binary synaptic layers in feature extraction, $\mathtt{SECA\text{-}II}(\cdot)$ combines the input currents

Table 2: Comparison with previous SoTA works on STBP and online training.

| Dataset | Method | Arch. | Param.(B) | Online | T | Acc.(%) |
|---|---|---|---|---|---|---|
| CIFAR-10 | STBP-tdBN (Zheng et al., 2021) | ResNet-19 | 50.48M | ✗ | 4 | 92.92 |
| | Dspike (Li et al., 2021) | ResNet-18 | 44.66M | ✗ | 4 | 93.66 |
| | TET (Deng et al., 2022) | ResNet-19 | 50.48M | ✗ | 4 | 94.44 |
| | GLIF (Yao et al., 2022) | ResNet-18 | 44.66M | ✗ | 4, 6 | 94.67, 94.88 |
| | SLTT (Meng et al., 2023) | ResNet-18 | 44.66M | ✔ | 6 | 94.44 |
| | **Ours** | **ResNet-18** | **2.82M** | **✔** | **4** | **95.51** |
| CIFAR-100 | Dspike (Li et al., 2021) | ResNet-18 | 44.84M | ✗ | 4 | 73.35 |
| | TET (Deng et al., 2022) | ResNet-19 | 50.57M | ✗ | 4 | 74.47 |
| | GLIF (Yao et al., 2022) | ResNet-18 | 44.84M | ✗ | 4, 6 | 76.42, 77.28 |
| | SLTT (Meng et al., 2023) | ResNet-18 | 44.84M | ✔ | 6 | 74.38 |
| | **Ours** | **ResNet-18** | **3.00M** | **✔** | **4** | **79.91** |
| ImageNet-200 | DCT (Garg et al., 2021) | VGG-13 | 38.02M+ | ✗ | 125 | 56.90 |
| | Online-LTL (Yang et al., 2022) | VGG-13 | 38.02M+ | ✔ | 16 | 54.82 |
| | Offline-LTL (Yang et al., 2022) | VGG-13 | 38.02M+ | ✗ | 16 | 55.37 |
| | ASGL (Wang et al., 2023) | VGG-13 | 38.02M | ✗ | 4, 8 | 56.57, 56.81 |
| | **Ours** | **VGG-13** | **2.77M** | **✔** | **4** | **60.68** |
| ImageNet-1k | STBP-tdBN (Zheng et al., 2021) | ResNet-34 | 87.12M | ✗ | 6 | 63.72 |
| | TET (Deng et al., 2022) | ResNet-34 | 87.12M | ✗ | 6 | 64.79 |
| | OTTT (Xiao et al., 2022) | ResNet-34 | 87.12M | ✔ | 6 | 65.15 |
| | SLTT (Meng et al., 2023) | ResNet-34 | 87.12M | ✔ | 6 | 66.19 |
| | **Ours** | **ResNet-34** | **7.40M** | **✔** | **4** | **68.07** |
| DVS-CIFAR10 | STBP-tdBN (Zheng et al., 2021) | ResNet-19 | 50.48M | ✗ | 10 | 67.80 |
| | Dspike (Li et al., 2021) | ResNet-18 | 44.66M | ✗ | 10 | 75.40 |
| | OTTT (Xiao et al., 2022) | VGG-SNN | 37.05M | ✔ | 10 | 76.30 |
| | NDOT (Jiang et al., 2024) | VGG-SNN | 37.05M | ✔ | 10 | 77.50 |
| | **Ours** | **ResNet-18** | **2.81M** | **✔** | **10** | **78.10** |
| | | **VGG-SNN** | **2.49M** | **✔** | **10** | **83.00** |

from both pre-synaptic and post-synaptic layers to further enhance the effectiveness of the attention mechanism. For ResNet backbone, in addition to downsampling convolutional layers, we usually consider **SECA-II**(·).

## 5 EXPERIMENTS

To validate the superiority of our proposed scheme compared to vanilla online learning framework, we investigate the learning performance of EM-PF model on various datasets with different data-scale and data-type, including CIFAR-10(100) (Krizhevsky et al., 2009), ImageNet-200(1k) (Deng et al., 2009) and DVS-CIFAR10 (Li et al., 2017). We comprehensively consider previous methods based on STBP and online training as our comparative works for ResNet (He et al., 2016; Hu et al., 2024b) and VGG (Simonyan & Zisserman, 2014) backbones. Training and implementation details have been provided in Appendix.

### 5.1 COMPARISON WITH PREVIOUS SoTA WORKS

**CIFAR-10 & CIFAR-100.** As shown in Tab.2, compared to tradition STBP and online learning framework, our scheme is based on floating-point spikes and binary synaptic weights, which saves approximately $15\times$ parameter memory, enabling effective online deployment for SNN models. In addition, we achieve higher learning precision within the same or fewer time-steps. For example, our method outperforms GLIF (Yao et al., 2022) and SLTT (Meng et al., 2023) with accuracies of 3.49% and 5.53% respectively on CIFAR-100, ResNet-18.

**ImageNet-200 & ImageNet-1k.** For large-scale datasets, our EM-PF model has also demonstrated significant advantages. For instance, we respectively achieve accuracies of 60.68% and 68.07% on

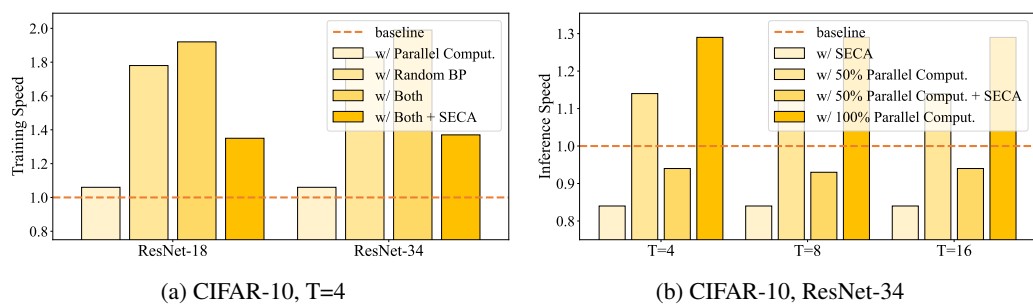

(a) CIFAR-10, T=4  (b) CIFAR-10, ResNet-34

Figure 4: Performance validation for random back-propagation and parallel computation.

Table 3: Parameter memory and accuracy of SNN models before and after utilizing SECA.

| Method | CIFAR-10, ResNet-18 | | CIFAR-100, ResNet-18 | | ImageNet-200, VGG-13 | |
|---|---|---|---|---|---|---|
| | Param.(B) | Acc.(%) | Param.(B) | Acc.(%) | Param.(B) | Acc.(%) |
| EM-PF model | 2.82M | 95.51 | 3.00M | 79.91 | 2.77M | 60.68 |
| **EM-PF model (+SECA)** | **2.82M** | **95.87 (+0.36)** | **3.00M** | **80.68 (+0.77)** | **2.77M** | **61.12 (+0.44)** |

ImageNet-200 and ImageNet-1k, which exceeds the corresponding online learning methods (Yang et al., 2022; Meng et al., 2023) within fewer time-steps and saves more than 90% of parameter memory.

**DVS-CIFAR10.** Our method can achieve effective information extraction for neuromorphic data. From Tab.2, one can note that our learning precision is 2.70% higher than Dspike (Li et al., 2021) and 5.50% higher than NDOT (Jiang et al., 2024) under the condition of utilizing the identical network backbone and time-steps.

## 5.2 VALIDATION STUDY FOR ACCELERATING COMPUTATION

As shown in Fig.4, we investigate the effects of random back-propagation and parallel computation on accelerating computation during the training and inference phases, respectively. According to Fig.4(a), directly adopting random back-propagation can increase the training speed by about 80%, while further combining parallel computation can increase the speed to over $1.9\times$. In the inference phase, when we choose the residual block shown in Fig.3(b), which means that 50% of the neurons will use parallel computing mode, the inference speed can be improved by about 15%. In the extreme case (100% parallel computation), the inference speed can be further improved to about $1.3\times$.

## 5.3 PERFORMANCE ANALYSIS FOR SECA

As shown in Tab.3, we explore the network performance before and after inserting SECA modules. One can note that SECA hardly introduces additional parameter memory and can provide extra precision improvement for binary synaptic layers. According to Fig.4, by combining random back-propagation and parallel computation, the online training speed based on SECA modules can even reach $1.3\times$ than that of vanilla online training. In addition, introducing parallel computation in the inference phase can also alleviate the problem of relatively slow inference speed in SNN models based on SECA to some extent.

## 6 CONCLUSIONS

In this paper, we systematically analyze the deficiencies of traditional online training, then propose a novel online learning framework based on floating-point spikes and binary synaptic weights, which effectively tackles the performance degradation problem caused by temporal dependent gradients and can achieve comprehensive model learning and deployment by flexibly combining various optimization techniques. Experimental results have verified that our proposed scheme can break through the limitations of previous methods and provide further inspiration for the future development of online learning.

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

# A APPENDIX

## A.1 PROOF OF THEOREM 4.2

**Theorem 4.2** *In the following two cases, the back-propagation of EM-PF model satisfies the condition of **Separable Backward Gradient** and $\forall i > t, \boldsymbol{\epsilon}^l[i,t] = \mathbf{0}$:*
*(i) $\boldsymbol{I}^l[1] \geq \theta^l[1] - \lambda^l[1]\boldsymbol{v}^l[0]; \forall t \geq 2, \boldsymbol{I}^l[t] \geq \theta^l[t] - \lambda^l[t]\theta^l[t-1]$.*
*(ii) $\forall t \in [1, T], \boldsymbol{I}^l[t] < \theta^l[t] - \lambda^l[t]\boldsymbol{v}^l[t-1]$.*

*Proof.* For case (i), EM-PF model will emit a spike at each time-step, which means that $\forall t \in [1, T], \frac{\partial \boldsymbol{s}^l[t]}{\partial \boldsymbol{m}^l[t]} = 1$. Combining with the definition of $\boldsymbol{\epsilon}^l[i,t]$, we will have:

$$
\begin{aligned}
\boldsymbol{\epsilon}^l[i,t] &= \frac{\partial \boldsymbol{s}^l[i]}{\partial \boldsymbol{m}^l[i]} \prod_{j=t+1}^{i} \frac{\partial \boldsymbol{m}^l[j]}{\partial \boldsymbol{m}^l[j-1]} \\
&= \mathbf{1} \cdot \prod_{j=t+1}^{i} \left( \lambda^l[j] + \frac{\partial \boldsymbol{m}^l[j]}{\partial \boldsymbol{s}^l[j-1]} \frac{\partial \boldsymbol{s}^l[j-1]}{\partial \boldsymbol{m}^l[j-1]} \right) \\
&= \mathbf{1} \cdot \prod_{j=t+1}^{i} \left( \lambda^l[j] - \lambda^l[j] \cdot \mathbf{1} \right) \\
&= \mathbf{0}.
\end{aligned}
\tag{S1}
$$

For case (ii), EM-PF model will keep silent at each time-step, which means that $\forall t \in [1, T], \frac{\partial \boldsymbol{s}^l[t]}{\partial \boldsymbol{m}^l[t]} = 0$. Combining with Eq.(S1), we will obviously conclude that $\forall i > t, \boldsymbol{\epsilon}^l[i,t] = \mathbf{0}$. □

## A.2 EXPERIMENTAL CONFIGURATION

For experimental cases in Tabs.2-3, we choose Stochastic Gradient Descent (Bottou, 2012) as our optimizer and Cosine Annealing (Loshchilov & Hutter, 2017) as our scheduler. The initial learning rate and weight decay are set to 0.01 and $5 \times 10^{-4}$, respectively. We consider various data augmentation techniques, including Auto-Augment (Cubuk et al., 2019), Cutout (DeVries & Taylor, 2017), and Mixup (Zhang et al., 2017). For ResNet backbone, we generally choose vanilla parallel computation block (version I plus version III), as shown in Fig.3(b). For VGG structure, we utilize the version of learnable parameters for ImageNet-200 and the version of membrane potential batch-normalization for DVS-CIFAR10. For experimental cases based on SECA, we all choose the EM-PF model with membrane potential batch-normalization (version II). More detailed experimental configuration has been provided in Tab.S1.

Table S1: Experimental setup for all training cases.

| Method | Arch. | SECA | Batchsize | Training Epochs | Version I | Version II | Version III |
|---|---|---|---|---|---|---|---|
| CIFAR-10 | ResNet-18 | ✗ | 64 | 300 | ✔ | - | ✔ |
|  |  | ✔ |  |  | - | ✔ | ✔ |
| CIFAR-100 | ResNet-18 | ✗ | 64 | 300 | ✔ | - | ✔ |
|  |  | ✔ |  |  | - | ✔ | ✔ |
| ImageNet-200 | VGG-13 | ✗ | 64 | 300 | ✔ | - | - |
|  |  | ✔ |  |  | - | ✔ | - |
| ImageNet-1k | ResNet-34 | ✗ | 256 | 120 | ✔ | - | ✔ |
| DVSCIFAR-10 | ResNet-18 | ✗ | 32 | 300 | - | ✔ | ✔ |
|  | VGG-SNN |  |  |  | - | ✔ | - |

## A.3 OVERALL ALGORITHM PSEUDO-CODE

**Algorithm 1** Online learning process for EM-PF model with various optimization techniques.

**Require:** SNN model $f_{\text{SNN}}(\boldsymbol{W}_{\text{float}}, \lambda, \theta, \alpha, \beta)$ with $L$ layers; Dataset $D$; Training time-steps $T$.
**Ensure:** Trained SNN model $f_{\text{SNN}}(\boldsymbol{W}, \lambda, \theta)$.
1: # Online Training
2: **for** (**Image**,**Label**) in $D$ **do**
3:   **for** $t = 1$ to $T$ **do**
4:     **for** $l = 1$ to $L$ **do**
5:       **if** Use the version of learnable parameters **then**
6:         EM-PF model performs forward propagation in Eq.(3)
7:       **else if** Use the version of membrane potential batch-normalization **then**
8:         EM-PF model performs forward propagation in Eq.(6)
9:       **else if** Use the version of parallel computation **then**
10:         $s^l[t] = \texttt{ActFunc}(\boldsymbol{I}^l[t] - \theta^l[t])$
11:       **end if**
12:       **if** Use SECA modules **then**
13:         Calculate and merge the channel attention scores through Eq.(10)
14:       **end if**
15:     **end for**
16:     **if** Use vanilla BP or the time of Random BP is equal to $t$ **then**
17:       Perform back-propagation as shown in Eq.(4) and update all learnable membrane-related parameters $\lambda[t], \theta[t], \alpha[t], \beta[t], \texttt{BN}_t(\cdot)$
18:     **end if**
19:   **end for**
20: **end for**
21: # Online deployment
22: **for** $l = 1$ to $L$ **do**
23:   Binarize the synaptic layer from $\boldsymbol{W}^l_{\text{float}}$ to $\boldsymbol{W}^l$
24:   **for** $t = 1$ to $T$ **do**
25:     **if** Use the version of membrane potential batch-normalization **then**
26:       Reparameterize $\alpha^l[t], \beta^l[t], \texttt{BN}_t(\cdot)$ into $\hat{\lambda}^l[t], \hat{\theta}^l[t], \hat{\boldsymbol{I}}^l[t]$ through Eq.(8)

27:       **end if**
28:    **end for**
29: **end for**
30: **return** $f_{\text{SNN}}(\boldsymbol{W}, \lambda, \theta)$.

