# OpenReview forum: "Comprehensive Online Training and Deployment for Spiking Neural Networks"
_ICLR.cc/2025/Conference — ICLR 2025 Conference Withdrawn Submission_

### Official Review · Reviewer_Z3bi · 2024-11-02

**Soundness:** 3
**Presentation:** 3
**Contribution:** 2
**Rating:** 5
**Confidence:** 3

**Summary:**

This paper proposes a novel online learning framework based on floating-point spikes and binary synaptic weights. It can tackle the performance degradation problem caused by temporal dependent gradients and can achieve comprehensive model learning and deployment.

**Strengths:**

The paper presents a well-motivated study and a novel solution. It thoughtfully considers both the training and testing phases, and the proposed parallel solution effectively enhances the efficiency of SNN processing. The writing is also clear and easy to follow.

**Weaknesses:**

My primary concern is whether the floating-point spike network qualifies as a true spiking neural network (SNN). According to Roy et al. (2019), "SNNs use the timing of signals (or spikes) to process information. Spikes are essentially binary events, either 0 or 1." In addition, SNNs are designed to mimic the information processing mechanisms in biological neural networks, typically involving a threshold-triggered mechanism for spike generation. In contrast, this paper employs ReLU activation. It seems that the model more like a binary neural network with a temporal component rather than a true SNN.

The proposed model, due to its small bit-width parameters, has a notably small parameter size. However, one of the advantages of SNNs is their inherent sparsity, which generally reduces computational demands and can enhance speed and throughput. While I understand that the parallel version can also improve speed, could the authors provide insights into the performance and efficiency of the other versions? How about the weight and activation sparsity of EM-PF model?

Roy et al. (2019), Towards spike-based machine intelligence with neuromorphic computing, Nature, 2019.

**Questions:**

The authors have proposed multiple versions of the EM-PF model, but it is unclear which specific version's results are presented in Table 2. Could the authors clarify which variant was used?

---

### Official Review · Reviewer_VHkb · 2024-11-02

**Soundness:** 3
**Presentation:** 3
**Contribution:** 3
**Rating:** 6
**Confidence:** 3

**Summary:**

The paper introduces the Efficient Multi-Precision Firing (EM-PF) model, building on the advantages of conventional online training while enhancing computational efficiency and reducing memory requirements. It combines spiking models with binary synaptic weights and floating-point spikes, which supports separate temporal gradients and full-stage optimization across both training and inference phases. The model demonstrates performance improvements on several benchmarks, including CIFAR-100 and ImageNet datasets, indicating its potential for broader applications in the field of spiking neural networks.

**Strengths:**

+ This work addresses significant limitations in the SNN domain and focues on enabling more efficient back-propagation through improved gradient separability.

+ Empirical results substantiate the effectiveness of the proposed methods, showcasing performance and memory savings on various datasets, supporting the paper's claims.

**Weaknesses:**

+ The introduction of membrane potential batch normalization and neuron-internal temporal parameters adds layers of complexity to the neuron model compared to traditional LIF-based SNNs. This likely results in increased training time costs. The paper should address this issue, including a comparative analysis of the time expenses associated with LIF-based BPTT in the discussion on computational efficiency.
+ While the use of binary weights to compress weight parameters is commendable, the paper lacks a comparative analysis of the impact of this compression on overall network performance before and after compression. Additional experiments discussing these effects would strengthen the understanding of the trade-offs involved.
+ The paper mentions acceleration in computation speed during the inference phases, but it lacks experimental data comparing the inference speed across different models and datasets. Including such a comparison, possibly in Table 2, would provide a clearer demonstration of the model's efficiency across various conditions and further validate the claimed improvements in computational speed.

**Questions:**

See Weaknesses.

---

### Official Review · Reviewer_dApA · 2024-11-04

**Soundness:** 1
**Presentation:** 2
**Contribution:** 1
**Rating:** 3
**Confidence:** 4

**Summary:**

The paper discusses EM-PF, a framework that claims to improve the efficiency of STBP towards attaining on-line training. It integrates in this framework ideas from past literature (and common practice in training SNNs) such as pruning the temporal back-propagation of gradient information across timesteps, or only selecting a subset of timesteps along which information is backpropagated, employing batch normalization and folding its parameters, propagating instead of binary spikes the delta of the membrane potential from the threshold, and increasing the parametrization of the neuron model to include decay and threshold as trainable parameters.

**Strengths:**

One likely strength of the paper is an attempt to formalize a unifying framework for describing different approaches/ideas for training SNNs, using back-propagation.

**Weaknesses:**

Reading the manuscript it feels that the authors have tried to implement everything they have read about SNNs so far in one framework and then have tried to describe it formally in a paper. However, I failed to identify any aspect of novelty, and what they describe as their contributions are to be really attributed to other works and innovations from past literature which they integrated into their implementation.

To be specific and exemplary of my remarks, for instance:
- the authors claim that EM-PF has (superior) backward gradient separability -- a fancy way of saying that the gradient is not (allowed to be) propagated backwards temporally only spatially thereby saving training memory -- by resorting to the ideas in the two references cited in l-051
- the authors introduce several variations of their EM-PF framework, which further amortizes efficiency, by deploying a set of methods from the literature they cite in l-103,104,360
- the authors report superior performance using their framework, after having practically converted their SNN model to an feed-forward ANNs namely propagating the (normalized) membrane potential and not binary spikes (and applying a few "tricks" from DNN literature like BN), and while testing them on spatial tasks, when most of the ramification to STBP regard the temporal processing part.

Most disappointing has been seeing a work that boasts about the superiority of SNNs in performance by simply adopting assumptions that convert SNNs to ANNs and eventually QNNs, and STBP to BP. When you remove the implicit self-recurrency by setting the decay to zero the model obviously is a simple integrator (IF). When you propagate the membrane and then reset it, the neuron becomes stateless. When you rectify the activation and use the straight through estimator as a gradient for it, then you have ReLU neurons, and suddenly you are in the DNNs world. Furthermore if you start to use BN and fold its parameters in the weights or the activations, while quantizing the weights, you are in QNNs.

**Questions:**

I believe the authors have mainly realized that the SNN neuron and network models are extremely versatile and expressive mathematically, so that they can emulate the ANN/DNN models (when in fact the latter have been derived originally by simplifying the SNN models functionally). If I have misinterpreted the assumptions made in this manuscript, and the authors can defend their innovations/novelty beyond that realization, or based on something I missed beyond the works that they have used from the literature, I would be willing to discuss and re-consider my assessment.

Next to that experiments need to be reported on temporal benchmarks not spatial, since the ramifications discussed involve the temporal domain (e.g. video processing, or speech, or other time-series, and without considering all data as one single input frame).

Finally, when claiming on-line learning, I expect to see a discussion and evaluation of the online and feasibility aspects; I dont see the online learning challenge as equivalent to weather training can fit in the memory of a GPU (this is an potential obstacle to overcome but not online learning per-se).

---

### Official Review · Reviewer_gbCc · 2024-11-04

**Soundness:** 2
**Presentation:** 3
**Contribution:** 2
**Rating:** 5
**Confidence:** 4

**Summary:**

This paper proposes a novel online learning framework based on floating-point spikes and binary synaptic weights, aiming to address the performance degradation issues of traditional online training methods. The authors propose Efficient Multi-Precision Firing (EM-PF) model which can effectively separate temporal gradients and achieve full-stage optimization. Experimental results demonstrate that the proposed method outperforms existing state-of-the-art techniques across multiple datasets, providing new insights for the future development of online learning.

**Strengths:**

The EM-PF model proposed in the paper effectively enhances learning capability and efficiency by integrating various optimization techniques, such as membrane potential batch normalization and SECA modules, demonstrating significant innovative value. By introducing SECA, the model improves classification accuracy while maintaining a low parameter count, showcasing excellent parameter utilization. The combination of random backpropagation and parallel computation leads to significant improvements in both training and inference speed, indicating the potential of this method for practical applications.

**Weaknesses:**

Is there an analysis of the energy consumption of the method?

**Questions:**

How well does the method work on deeper SNNs.

---

### Note · Authors · 2024-11-13

I have read and agree with the venue's withdrawal policy on behalf of myself and my co-authors.